# Regression of *Schistosoma mansoni* associated morbidity among Ugandan preschool children following praziquantel treatment: A randomised trial

**Allen Nalugwa** [1]*, **Edridah Muheki Tukahebwa**[2], **Annette Olsen**[3], **Fred Nuwaha**[4]

**1** Child Health and Development Centre, College of Health Sciences, Makerere University, Kampala, Uganda, **2** Neglected Tropical Diseases, Vector Control Division, Ministry of Health, Kampala, Uganda, **3** Parasitology and Aquatic Diseases, Faculty of Health and Medical Sciences, University of Copenhagen, Copenhagen, Denmark, **4** Disease Control and Prevention, Makerere University, Kampala, Uganda

* allennalugwa@yahoo.co.uk

**Data Availability Statement:** All relevant data are within the manuscript and its Supporting information files.

## Abstract

Preschool children suffer from morbidity attributable to *Schistosoma mansoni*. We compared a single and double dose of praziquantel treatment on the regression of *S. mansoni* associated morbidity in children less than six years in Uganda. We measured the sizes of spleen and liver as well as liver fibrosis before treatment and 8 months after treatment among children who either received one dose (n = 201) or two doses (n = 184) of praziquantel (standard oral dose of 40 mg/kg body weight). Heamoglobin measurements were also taken. Overall, liver enlargement reduced from 52.2% (95% CI (Confidence interval) 45.1, 59.3) to 17.9% (95% CI 12.9, 23.9) with a single dose and from 48.4 (95% CI 40.9, 55.8) to 17.9% (95% CI 12.7, 24.3) with a double dose and there was no significant difference between the changes in proportion of children with enlarged liver between the two treatment groups. The proportion of children with enlarged spleen was not significantly reduced in the group treated with either one or two doses, 47.8% (95% CI 41.7, 54.9) to 45.3% (95% CI 38.3, 52.4) and 48.4% (95% CI 40.9,55.8) to 40.8% 95% CI 33.6, 48.2), respectively. Liver fibrosis detected among children getting single dose (n = 9) or double doses (n = 13) resolved after treatment with praziquantel. The number of children with low heamoglobin significantly reduced from 51.2% (95% CI 44.1, 58.3) to 0.5% (0.2, 0.8) and 61.4% (95% CI 53.9,68.5) to 1.1% (95% CI 0.1, 3.9) after single and double dose treatment, respectively. These results suggest that there is no evidence of a difference in effect between one dose of praziquantel and two doses in reversing morbidity attributable to *S. mansoni* among children less than six years of age.

## Introduction

Preschool children (PSC) less than six years old get infected [1–5] with *Schistosoma mansoni* and develop associated morbidity such as anaemia and hepatomegaly [6–10]. With heavy

**Funding:** This study was supported by the Danish Ministry of Foreign Affairs-Project No. 09-100KU. The funders had no role in the study design, data collection and anaysis, decision to publish, or preparation of the manuscript.

**Competing interests:** The authors have declared that no competing interests exist.

infection there is a higher likelihood that the hepatomegaly may progress to hepatic fibrosis even at this early age [6, 8]. Even with these observations, young children less than 94 cm in height (about ≤ 4 years of age) are not included in community and or school-based treatment programmes for schistosomiasis [11]. Several reasons preclude treatment of the young children. First is the ostensive belief that complications of chronic schistosomiasis takes a long time to develop and therefore young children can wait [6, 12]. Second is the lack of child friendly praziquantel for treatment of children [13, 14]. The current tablet of praziquantel is big and bitter with the wherewithal to cause choking and vomiting among young children. Third, determination of the praziquantel dose within this age group using height is a herculean task and is considered rather inaccurate [15]. Finally, because organisation of mass drug delivery for schistosomiasis outside the school system is a logistical nightmare in most of sub-Saharan Africa, pre-school-aged children and out of school children are rarely treated [1, 6, 7]. We and others have previously characterised infection and morbidity associated with *S. mansoni* in children less than six years and demonstrated that treatment with praziquantel was safe and efficacious [16–19]. This is a follow-up report to document the long-term effects of such treatment on morbidity associated with intestinal schistosomiasis. In this study, we assessed the impact of praziquantel on *S. mansoni* associated morbidity eight months after initial treatment among PSC in Uganda, a country where about one in four people are estimated to be infected with *S. mansoni* [2]. This study, therefore, was intended to determine whether *S. mansoni*-associated morbidity in PSC regressed after treatment with PZQ, comparing one versus two doses. The information generated is helpful in guiding policy and practice decisions regarding the routine treatment of this important age group as the goal of schistosomiasis control moves toward elimination [20].

## Materials and methods

### Study area and participants

This paper is a continuation of a previous clinical trial, details of study methods described in these reports [3, 9, 17] but will be summarized here. The study was carried out in 26 *S. mansoni* endemic fishing communities along Lake Victoria shoreline in eastern Uganda in the districts of Bugiri, Buikwe, Jinja, Mayuge and Namayingo. Children aged 12 to 60 months, who were tested egg positive, were treated with either a single or double dose of PZQ, and were further investigated for *S. mansoni* associated morbidity parameters before (June 2013) and 8 months after (February, 2014) chemotherapy. This study is registered with ClinicalTrial.gov.

### Measures

The study procedures included stool parasitological examination, anaemia assessment and abdominal ultrasound examination as described below.

**Parasitological examination.** Following community sensitization on the ongoing study and written consent, caregivers whose children were to participate in the study were given orientation on how to handle and submit the stool samples of their children. Stool containers (polythene sheets) labelled with the child's identification number and name were given out to the respective parents. For each child one stool sample was collected on three consecutive days; multiple stool collections were proposed due to day-to-day variation in egg counts of *S. mansoni* [21, 22]. The Kato–Katz technique was used to prepare stool smears on slides for microscopic examination [23]. Two slides were prepared and examined for each sample; totalling six slides for each child. A small amount of faeces was pressed through a steel screen to remove large debris, the sieved stool filled into a 41.7 mg hole in a template placed on a slide. The specimen on the slide was covered by a piece of cellophane soaked in glycerol with

malachite green used as a cover slip. The two faecal smears were each examined under a microscope (100x magnification) and eggs on each slide were counted and recorded by two different experienced field technicians. To ensure the accuracy of the egg counts a 10% of the slides from each field technician were chosen at random and re-read by a senior technician. In case of discrepancies, the slides were re-read and consensus obtained. The outcome infection intensities were classified according to WHO guidelines [24] as light: 1-99EPG, moderate:100-399EPG and heavy: ≥400EPG. Intensity of infection (expressed in eggs per gram of stool, EPG) was calculated by multiplying the mean for the six slides (two slides for each of the three stool samples) by a factor of 24 [25].

**Anaemia assessment.** A finger-prick blood sample was taken from all the recruited children, and their heamoglobin (Hb) concentration (g/dL) was measured using a portable Hemo-Cue® photometer (Ängelholm, Sweden). Anaemia was defined as absent (Hb = >11 g/dL), mild (Hb = 8–10.9 g/dL), moderate (Hb = 6–7.9 g/dL) or severe (Hb<6 g/dL) according to the World Health Organization guidelines [26].

**Abdominal ultrasound examination.** Abdominal ultrasonography was performed using a portable ultrasonographical device (Aloka, SonocameraSSD-500 Tokyo, Japan) with a convex 3.5 MHz transducer, according to WHO standard guidelines [27]. The children were examined lying on their backs with their legs stretched on an examination table. Measurements from the upper to the caudal margin in the left parasternal line (PSL) were done for the spleen length (SL) and the left liver lobe. The size of the right liver lobe was measured in the right mid-clavicular line (MCL), whereas the portal-vein diameter (PVD) was measured at the point where the portal vein enters the porta hepatica at the lower end of the caudate lobe. Height-related reference data from a healthy Senegalese community [27, 28] were used to measure liver and spleen size. The values for organ sizes:, spleen length, left liver lobe (PSL) and right liver lobe (MCL) were then classified as 'normal' if they were below or equal to the mean + 2 SD; 'moderately enlarged' if they were more than 2 SD but below or equal to the mean + 4 SD, and 'severely enlarged' if they were above the mean + 4 SD. In children with texture patterns suggestive of periportal fibrosis (PPF), a picture was taken and the corresponding texture pattern score recorded. Liver texture patterns A and B were considered to show no liver fibrosis while C, D, E, and F indicated disease. In addition, the age, sex, and district were recorded.

## Treatment

The infected children were individually randomized (by an independent statistician) to single or double dose PZQ treatment groups by computer random number generation in Stata/IC 12.0. Each child infected with *S. mansoni* was treated according to their weight with PZQ (40 mg/kg body weight) based on standard protocols to ensure adherence and mitigation of side effects [17–19]. The tablets were crushed [16] and mixed with drinking water to facilitate oral uptake in small children [29]. The children were given a piece of bread before drug administration, and orange juice was provided after drug administration to minimize gastrointestinal side effects [13, 30, 31] and mask the sour taste of PZQ [32], respectively. Treatment was performed by an experienced nurse in the presence of each child's caretaker. The children were also each given a tablet of albendazole (Alzental 400 mg). Two weeks after the first treatment, children who were randomized into the double dose group were given a second dose of PZQ. Treated children remained under supervision for a period of 30 minutes to monitor any immediate reactions as a result of drug administration and the caregivers were further encouraged to report any treatment-related symptoms observed in the children 24 hours post-treatment. In case of adverse events, the affected children would be referred to the nearest local health services.

**Sample size.** The study compared the effect of two different PZQ treatment strategies (single dose versus double dose) on morbidity regression. It was expected that 8 months after treatment, morbidity as measured by liver size would regress by a difference of at least 12% in the two groups (75% in the two-dose group and about 63% in the one dose group). Using the formula of comparison of two proportions [21] with 90% power and at the 5% level of significance the sample size calculated was at least 343 children in each group after adjustment for a 25% loss to follow-up.

## Data analysis

Statistical analysis was carried out in Stata 14 (StataCorp; College Station, TX, USA) in conjunction with SPSS 23 (IBM, New York). To detect a change between follow up and baseline variables, categorical data were compared using the McNemar-Bowker test, and Paired *t*-test was used to compare continuous variables. Statistical significance was defined by a *P* value less than 0.05. To test whether the double dose was different from the single dose in influencing morbidity indices, we compared for a change in parameters before treatment and at follow up within the two groups using the Chi-square test statistic with Yates correction.

## Ethical review

Ethical approval and clearance were obtained from Makerere University, Ethics Committee and the Uganda National Council for Science and Technology (Ref. No. UNCST-HS1274), respectively. The trial was registered on May 30, 2013 with ClinicalTrial.gov, identifier: NCT01901484. The aim and procedure of the study were explained to the caregivers of the recruited children in the local language (Lusoga), and assent obtained before PZQ chemotherapy. Only those children who had written informed consent from their caregivers were included in this study.

# Results

Of a total of 686 children who received treatment, 385 were successfully retraced at the 8 months follow-up (201 in the single dose group and 184 in the double dose group) and only these were included in further analysis (Fig 1).

The children who received one dose were similar to the children who received double dose in terms of age sex and infection intensity (Table 1), at eight months follow-up.

The children lost to follow-up (301) and those retained (385) in the study were comparable in terms of age, sex and intensity of infection (Table 2).

## Morbidity regression

Morbidity reported here refers to anaemia, liver and spleen sizes as well a liver image patterns including progressive hepatic fibrosis. All children had normal sizes of hepatic portal vein at baseline and at follow-up and the results of portal vein diameter (PVD) are therefore not reported. Table 3 summarizes sonographic observations of morbidity regression detected after treatment with the two different PZQ doses at the two time points; baseline and 8 months post treatment.

The percent of children with normal spleen (P = 0.87) and liver (P = 0.24) at baseline were similar for those who received single dose compared to those who received double dose.

The proportion of children with enlarged spleen, presented as splenic length, was not significantly reduced in any of the two PZQ treatment groups as the percentage changed from 47.8% (95% CI 41.7, 54.9) to 45.3% (95% CI 38.3, 52.4) in the single dose group and from

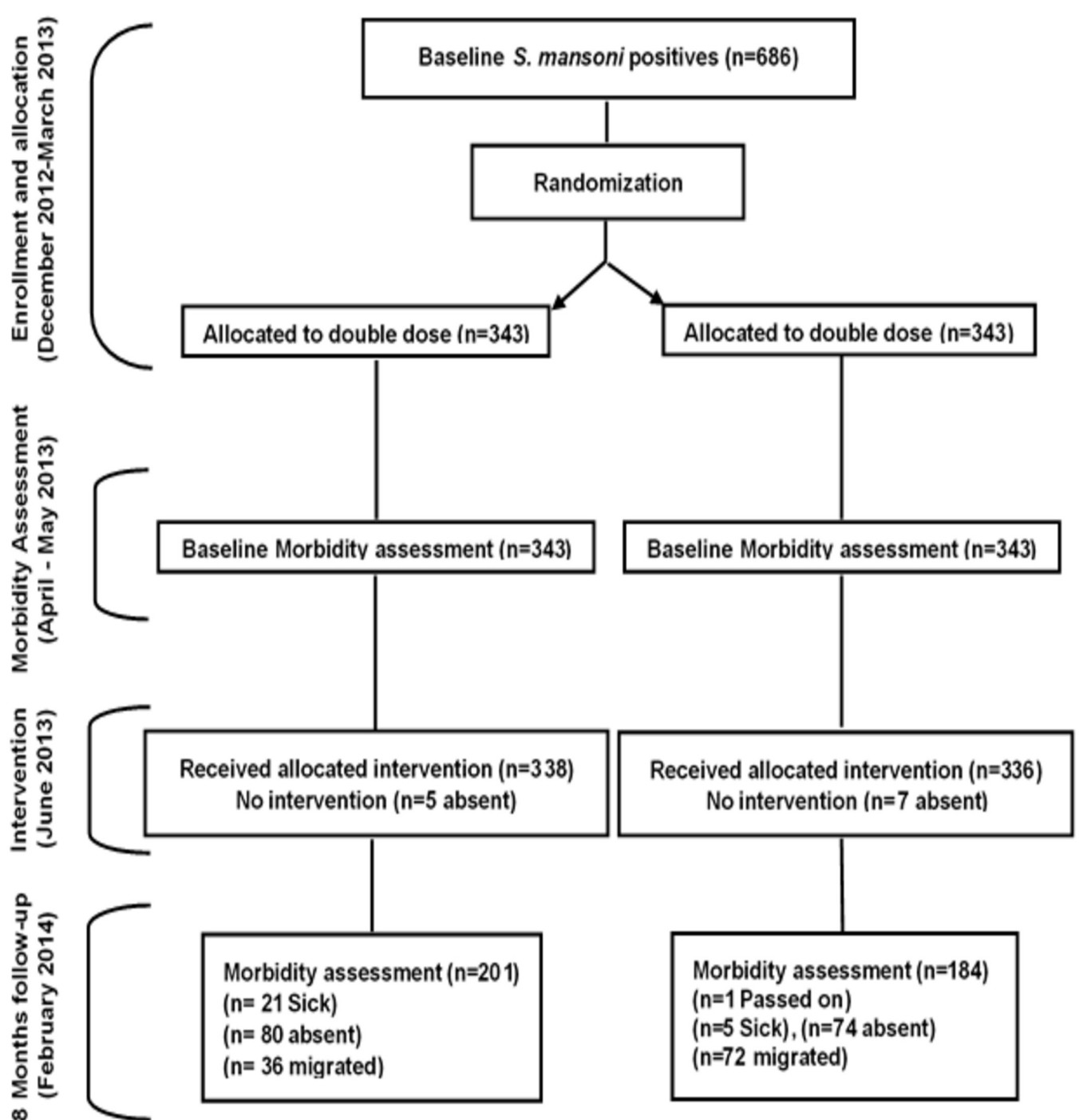

**Fig 1. Enrolment, randomization, follow-up, and inclusion in the final analysis comparing between two treatment groups.**

48.4% (95% CI 40.9,55.8) to 40.8% 95% CI 33.6, 48.2), in the double dose group. The much-enlarged liver lobe measured as PSL (left parasternal line) significantly ($P < 0.005$) reduced from 10.9% to 0.5% and from 8.7% to 1.6% after single and double dose treatment, respectively. Overall, liver enlargement reduced from 52.2% (95% CI 45.1, 59.3) to 17.9% (95% CI 12.9,23.9) with a single dose and from 48.4 (95% CI 40.9, 55.8) to 17.9% (95% CI 12.7, 24.3)

**Table 1. Comparison of preschool children at 8 months (N = 385) with regard to type of treatment, age, sex and infection intensity.**

| Variable | One Dose n (%) 201 (100) | Two doses n (%) 184 (100) |
|---|---|---|
| **Sex** | | |
| Female | 111(55.2) | 80 (43.5) |
| Male | 90 (44.8) | 104 (56.5) |
| **Age in months** | | |
| ≥ 24 | 7 (3.5) | 4 (2.2) |
| 25–36 | 42 (21) | 36 (19.6) |
| 37–48 | 61 (30.3) | 57 (31) |
| 49–60 | 91 (45.2) | 87 (47.2) |
| **Intensity of infection baseline (epg)** | | |
| 1–99 | 121 (60.2) | 103 (56) |
| 100–399 | 47 (23.4) | 40 (21.7) |
| ≥ 400 | 33 (16.4) | 41 (22.3) |

**Table 2. Comparison of preschool children lost to 8 months follow-up (n = 301) and those that were retained in the study (n = 385) with regard to age, sex and infection intensity.**

| Characteristic | Lost to follow-up n (%) 301 (100) | Retained in study n (%) 385 (100) |
|---|---|---|
| **Age group (months)** | | |
| 12–24 | 13 (4.3) | 13 (3.4) |
| 25–36 | 54 (17.9) | 75 (19.5) |
| 37–48 | 92 (30.6) | 118 (30.6) |
| 49–60 | 142 (47.2) | 179 (46.5) |
| **Sex** | | |
| Female | 153 (50.8) | 191 (49.6) |
| Male | 148 (49.2) | 194 (50.4) |
| **Infection intensity (EPG)** | | |
| Low/light (1–99) | 176 (58.5) | 224 (58.2) |
| Moderate (100–399) | 70 (23.3) | 87 (22.6) |
| Heavy (≥ 400) | 55 (18.2) | 74 (19.2) |

with a double dose. There was no significant change in the right liver lobe size among children treated with single dose compared to those treated with a double dose. The change in liver and spleen sizes were the same among children treated with single dose and those treated with a double dose ($P > 0.05$).

## Liver image patterns and progressive hepatic fibrosis

Liver texture patterns B1, B2, C1 and C2 presented in some children at baseline were completely resolved by both treatments at the 8 months follow up (Table 4).

There was no difference between single dose and double dose regarding normalisation of liver texture patterns ($P = 0.30$).

## Anaemia

At 8 months follow-up there was a significant increase ($P < 0.001$) in normal Hb values as shown in Table 5. Proportion of children with normal Hb increased from 51.2% (95% CI 44.1,

**Table 3. Liver and spleen size between single and double dose treatments before and after treatment with praziquantel.**

| Characteristic | Single dose (n = 201) | | | Double dose (n = 184) | | |
|---|---|---|---|---|---|---|
| | Pre-treatment | Post-treatment | Pre vs Post treatment | Pre- treatment | Post- treatment | Pre vs Post treatment |
| | n (%) | n (%) | *P*- value | n (%) | n (%) | *P*- value |
| **Spleen length (SL)** | | | | | | |
| Normal (≤ +2SD) | 105 (52.3) | 110 (54.7) | ns | 95 (51.6) | 109 (59.2) | ns |
| Moderate enlargement (> +2 ≤ +4SD) | 63 (31.3) | 62 (30.9) | | 60 (32.6) | 46 (25) | |
| Severe enlargement (≥+4) | 33 (16.4) | 29 (14.4) | | 29 (15.8) | 29 (15.8) | |
| **Liver PSL** | | | | | | |
| Normal (≤ +2SD) | 96 (47.8) | 165 (82.1) | <0.001 | 95 (51.6) | 151 (82.1) | < 0.001 |
| Moderate enlargement (> +2 ≤ +4SD) | 83 (41.3) | 35 (17.4) | | 73 (39.7) | 30 (16.3) | |
| Much enlarged (≥+4) | 22 (10.9) | 1 (0.5) | | 16 (8.7) | 3 (1.6) | |
| **Liver MCL** | | | | | | |
| Normal (≤ +2SD) | 201 (100) | 201 (100) | ns | 183 (99.5) | 184 (100) | ns |
| Moderately shrunk (> +2 ≤ +4SD) | 0 (0) | 0 (0) | | 1 (0.5) | 0 (0) | |
| Severely shrunk (≥+4) | 0 (0) | 0 (0) | | 0 (0) | 0 (0) | |

ns: not significant at P ≥ 0.05 SD-Standard deviation.

**Table 4. Comparison of liver texture pattern between single and double dose before and after treatment with praziquantel.**

| Characteristic | Single Dose | | Double Dose | |
|---|---|---|---|---|
| | Pre-treatment-n (%) | Post-treatment-n (%) | Pre-treatment- n (%) | Post-treatment-n (%) |
| **Liver texture pattern (LTP)** | | | | |
| Normal (A) | 183 (91) | 198 (98.5) | 159 (86.4) | 183 (99.5) |
| Feather streaks (B) | 9 (4.5) | 3 (1.5) | 12 (6.5) | 1 (0.5) |
| Flying saucers (B1) | 4 (2.0) | 0 | 5 (2.7) | 0 |
| Spider thickening (B2) | 2 (1.0) | 0 | 8 (4.4) | 0 |
| Peripheral rings (C1) | 1 (0.5) | 0 | 0 | 0 |
| Prominent pipe stems (C2) | 2 (1.0) | 0 | 0 | 0 |

**Table 5. Comparison of heamoglobin between single and double dose treatments before and after treatment with praziquantel.**

| Haemoglobin (g/dL) | Single dose | | | Double dose | | |
|---|---|---|---|---|---|---|
| | Pre-treatment | Post-treatment | p-value | Pre-treatment | Post-treatment | p-value |
| Normal (>11) | 98 (48.8) | 192 (95.5) | <0.001 | 71 (38.6) | 182 (98.9) | <0.001 |
| Mild anaemic (8–10.9) | 46 (22.9) | 5 (2.5) | | 53 (28.8) | 1 (0.5) | |
| Moderate anaemic (6–7.9) | 54 (26.9) | 4 (2.0) | | 54 (29.4) | 0 (0) | |
| Severe anaemic (< 6) | 3 (1.4) | 0 (0) | | 6 (3.2) | 1 (0.5) | |

58.3) at baseline to 95.5% (95% CI 91.7, 97.9) in the single dose group and from 38.6% (95% CI 31.5, 45.6) to 98.9% (95% CI 96.1, 99.9) in the double dose group. The percent mean increases in haemoglobin (follow-up minus baseline Hb divided by baseline Hb) was the same among children who received one dose compared to those who received two doses (One-way ANOVA, F-ratio, 1.037, one degree of freedom, *P* = 0.9).

## Discussion

The study compared the effectiveness of a single contrasted with a double dose of PZQ in reversing *S. mansoni* associated morbidity in children aged 1–5 years and it clearly shows that a single dose was as effective as a double dose in reduction of the left size of the liver lobe, normalisation of liver image patterns, regression of peri-portal fibrosis and in increasing haemoglobin. Neither single nor double dose PZQ were effective in reducing the size of the spleen in this population.

To our knowledge, this is the first study to follow-up children less than six years infected with *S. mansoni* and treated with PZQ for up to 8 months. In a study in Kenya among children follow-up period was up to 4 months and the only morbidity parameter that improved was the proportion of children without anaemia [6, 7]. Splenomegaly, left lobe hepatomegaly and children with image patterns of type B [IPB] increased instead. Possible reasons for these differences compared to our study could be due to: a much higher cure rate attributable to PZQ in our study compared to the Kenyan study [6, 7, 16], increased in malaria prevalence observed in the Kenyan study between baseline and follow-up [6, 7], and differential restricted classification of hepatic pathology into normal versus IPB as well as relatively shorter time of follow-up [33–35].

Our results are however similar to what was observed in Zambia [35] and the Sudan [33] among older children where the proportion of children with grossly enlarged livers significantly reduced. In Zambia the reduction was from 18% to 6.8% 14 months after treatment and in Sudan the reduction was from 27% to 4.6% 23 months after anti-schistosomal therapy. All the pre-school children in our study with abnormal liver texture patterns including those with progressive hepatic fibrosis [image pattern C] reverted to normal after treatment with PZQ. Though the numbers are small these results of our study could suggest that treatment at an early age has a higher likelihood of reversing liver abnormalities and has the wherewithal of completely halting progression of liver pathology associated with *S. mansoni* [36]. The fact that liver rather than splenic size reduced after treatment with PZQ indicates that *S. mansoni* infections could be responsible for much of the liver morbidity observed in preschool children in this study. We had earlier observed that the size of the left liver lobe, but not the right liver lobe and the spleen, are correlated with intensity of infection with *S. mansoni* among children aged less than six years [9]; Other studies [6] also indicate that enlargement of the left liver lobe but not the spleen, is a pattern related to infection with *S. mansoni* in children less than six years as well as in older children [33–35].

There was no change in the sizes of the right lobe of the liver as well as of the spleen. All the preschool children included in this study presented with normal right lobe of the liver at baseline and through the eight months follow-up. On the other hand, although over 45% of the children presented with abnormally enlarged spleens at baseline, this did not change following PZQ treatment. The persistent spleen enlargement observed in our study and what has been reported in other studies among preschool children [6] as well as school going children after treatment with PZQ may be attributed to other causes other than *S. mansoni* e.g. malaria infection. Malaria is a leading cause of morbidity in children in sub Saharan Africa [37] and spleen enlargement is associated with intensity of transmission [38]. Moreover, malaria and intestinal schistosomiasis are co-endemic in SSA [39] and both have pathological effects on the spleen [40]. The marked increase in Hb after treatment in this study is difficult to interpret. The likely explanation is that anaemia in this study was related with *S. mansoni* infections [17] and if so, we can say that decrease in the infections at follow-up led to increased Hb as previously observed elsewhere [10]. However, 9 children (2.3%), who were found severely anaemic, were given iron tablets on top of the PZQ treatment and this may contribute slightly to the increase

in Hb. Other common causes of anaemia in Uganda include infections with hookworm and malaria. Malaria prevalence was not assessed in this study and all children in the study were given albendazole although less than 5% were infected with hookworm [3, 17]. Thus, the effects of albendazole and iron treatment could not solely explain the increase in heamoglobin.

## Conclusions

We conclude that hepatomegaly, anaemia as well as liver fibrosis in preschool children infected with *S. mansoni* regressed after PZQ chemotherapy. There was no difference between single dose and double dose PZQ in reduction of the *S. mansoni* associated morbidity. Thus, our data suggest that preschool children benefit extensively from treatment and should be targeted for treatment in schistosomiasis control programmes. It is recommended that treatment programmes should adopt the use of one dose PZQ, which is equally effective as two doses in reversing schistosomiasis related morbidity. Administration of one dose is more cost-effective than two doses and in addition, it is more operationally feasible.

## Limitations of the study

The loss to follow up was high but this was expected in schistosomiasis endemic regions where migration is very common [41]. Nevertheless, we had enough numbers at 8 months to give us adequate power to ($> 0.8$) compare changes in liver size assuming a difference of 12% for single dose compared to double dose. Besides, this study compared parameters of the same children at baseline and at follow-up. Therefore, the conclusions of the study still remain relevant as children lost to follow-up are not included in estimating comparator parameters at baseline. A second limitation of the study is that infection with malaria was not assessed. Malaria can modify morbidity associated with *S. mansoni*. Although malaria transmission in the study area is perennial and stable with about 40% of children less than five years infected, seasonal spikes occasionally occur [42]. Finally, for ethical reasons it was not possible to have a control group of children that could not be treated.

## Supporting information

**S1 File. Data file.**
(SAV)

**S2 File. Research registration clearance and protocol.**
(RAR)

## Acknowledgments

We sincerely thank the parents and preschool children who participated in this study. We are grateful to the Vector Control Division in Uganda for the field equipment and dedicated field technicians. I am grateful to the co-authors for the helpful discussions.

## Author Contributions

**Conceptualization:** Allen Nalugwa, Edridah Muheki Tukahebwa, Annette Olsen, Fred Nuwaha.

**Data curation:** Allen Nalugwa.

**Formal analysis:** Allen Nalugwa, Fred Nuwaha.

**Investigation:** Allen Nalugwa.

**Methodology:** Allen Nalugwa.

**Supervision:** Allen Nalugwa, Edridah Muheki Tukahebwa, Annette Olsen, Fred Nuwaha.

**Validation:** Allen Nalugwa, Edridah Muheki Tukahebwa, Annette Olsen, Fred Nuwaha.

**Visualization:** Allen Nalugwa.

**Writing – original draft:** Allen Nalugwa, Annette Olsen, Fred Nuwaha.

**Writing – review & editing:** Allen Nalugwa, Edridah Muheki Tukahebwa, Annette Olsen, Fred Nuwaha.

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
