## [Decision Letter · Decision Letter 0]

30 Jun 2020

PONE-D-20-12223

Regression of Schistosoma mansoni associated morbidity among Ugandan preschool-aged children following praziquantel treatment: a randomized controlled trial

PLOS ONE

Dear Dr. Nalugwa,

Thank you for submitting your manuscript to PLOS ONE. After careful consideration, we feel that it has merit but does not fully meet PLOS ONE’s publication criteria as it currently stands. Therefore, we invite you to submit a revised version of the manuscript that addresses the all of the points raised by the Reviewers during the review process.

We look forward to receiving your revised manuscript.

Kind regards,

David Joseph Diemert, M.D.

Academic Editor

PLOS ONE

Journal Requirements:

2. PLOS ONE requires that methods are described in enough detail to allow suitably skilled investigators to fully replicate the study. See https://journals.plos.org/plosone/s/submission-guidelines#loc-materials-and-methods for more details. Please provide more details on the methods described under the 'Measures' subsection of your manuscript.

Please also provide a sample size calculation.

3. Thank you for stating in your ethics statement "Only those children who had written informed consent from their caregivers were included."

Please include this in the text of your manuscript as well. In addition, thank you for stating in the text of your manuscript that "Ethical approval and clearance were obtained from Makerere University, Ethics Committee and the Uganda National Council for Science and Technology (Ref. No. UNCST-HS1274), respectively. The trial was registered with ClinicalTrial.gov, identifier: NCT01901484". Please include this in your ethics statement as well.

Finally, we note that your study was registered on May 30, 2013. Please add this information to the text of your manuscript. In addition, please also provide the date range in which this study took place in the text of your manuscript.

Additional Editor Comments (if provided):

Reviewers' comments:

Reviewer's Responses to Questions

**Comments to the Author**

1. Is the manuscript technically sound, and do the data support the conclusions?

Reviewer #1: Partly

Reviewer #2: Partly

Reviewer #3: Yes

Reviewer #4: Partly

Reviewer #5: Yes

2. Has the statistical analysis been performed appropriately and rigorously? 

Reviewer #1: Yes

Reviewer #2: Yes

Reviewer #3: Yes

Reviewer #4: Yes

Reviewer #5: Yes

3. Have the authors made all data underlying the findings in their manuscript fully available?

Reviewer #1: Yes

Reviewer #2: Yes

Reviewer #3: Yes

Reviewer #4: No

Reviewer #5: Yes

4. Is the manuscript presented in an intelligible fashion and written in standard English?

Reviewer #1: Yes

Reviewer #2: Yes

Reviewer #3: Yes

Reviewer #4: Yes

Reviewer #5: Yes

5. Review Comments to the Author

Reviewer #1: The reason preschool age children are not typically included in mass drug administration (MDA) campaigns is not because the drug itself is potentially toxic to this age group but because the pill size is so large that there is a choking hazard. Thus, the authors' characterization of the reluctance to treat this age group in lines 88-90 is misleading and should be corrected. Along these same lines, it would be important to know whether the children in this study were treated with whole or crushed pills. This information is not provided and should be in the material and methods.

A major limitation of the study is the failure to assess malaria in the study participants. Malaria in children is associated with both image pattern B and anemia. Thus, all the study results are potentially confounded by seasonal malaria, especially as the follow up was conducted at 8 months rather than a year after baseline. In the very least, the authors should describe the prevalence of malaria in this region, its typical yearly timing in relation to the baseline and follow up of this study, and describe any malaria interventions that were conducted during this period. In addition, they should cite literature on the association of image pattern B with malaria and note that in many studies, image pattern B, while not normal, is not considered to be real schistosomiasis-associated fibrosis. Only image pattern C or greater is usually included in ultrasound studies of Schistosoma mansoni.

Reviewer #2: General: this is a simple, useful study of antiparasitic drug administration in a neglected subpopulation and will be a useful contribution to public health policy development in this area. There are several major questions that need to be addressed to clarify its contribution.

Title: While randomized, I don’t believe this can be considered a controlled trial since there is no control (untreated) group. Consider something like “a randomized dosing strategy trial” or “a randomized trial of one- vs. two-dose treatment”.

General: there is some confusion in language with regard to claims about morbidity. Egg-positive stool and egg counts are markers of infection/intensity, not morbidity (e.g. line 93). Even isolated hepatomegaly is a bit dubious as ‘morbidity’ – it’s more of just a concerning physical finding, unless you clearly link it to later end-organ functional damage and symptoms. If you do choose to keep using the term ‘morbidity’ to refer to asymptomatic liver enlargement, consider explicitly stating that you are treating hepatosplenomegaly as a form of morbidity even if isolated.

Introduction and throughout: the definition of preschool as ‘under 6 years’, while correct in Uganda, will complicate the attempt to generalize these findings to many other countries in subSaharan Africa that start school earlier. I believe e.g. ‘under 5’ is a more common cutoff. This is especially true as nearly half the study sample is children between 5 and 6 years – ie children who in most other SSA settings would in fact be school-aged and already eligible for praziquantel MDA. This is then further complicated by the fact that technically WHO recommendations are to administer PZQ to children based on a height minimum of 94cm (which is the WHO 50th growth percentile for children aged about 2.5 years), so virtually all of the study sample would in fact have been PZQ-eligible (if able to swallow pills, in the absence of a liquid formulation) during a door-to-door MDA. Essentially, authors need to clarify the situation in Uganda and what age groups are actually being currently excluded. Please consider that this would ideally include:

- In the intro: telling readers: in Uganda, are PZQ MDAs exclusively done in schools (where they might only reach school-aged kids) or also door-to-door? Is there any coverage data available from previous campaigns to provide a sense of what fraction of young children by age are actually getting PZQ in MDAs? Or if Uganda has a national policy actively excluding children under 6, please state and cite, but recognize this conflicts with WHO recommendations.

- In the analysis: it may be worthwhile to run a subgroup analysis restricted to whatever age subgroup genuinely has very low MDA coverage. Understanding this might be underpowered, findings could still be suggestive, and contrary findings (e.g. minimal starting hepatomegaly, or no change with treatment) would require discussion.

- In the discussion: What are the implications of the above issues for your recommendations, both for Uganda and for other SSA countries where children above 5 years are generally already included in existing campaigns?

Introduction: Currently, authors are primarily using the introduction to make the cases that preschool-aged children should be treated at all, but then the study focuses on frequency of treatment instead. We need additional text to make the case to the reader of why the study actually performed adds value. I.e., was there data or concern suggesting that younger kids were likely to require a double dose and thus would not benefit from inclusion in MDAs (which to my knowledge are only single-dose)?

Introduction: Given that your primary finding is going to be improvement in hepatomegaly, it would be helpful to more clearly link duration of hepatomegaly to poor subsequent clinical outcomes here if you can. Otherwise a reader might wonder how important it really is to resolve asymptomatic hepatomegaly at e.g. age 5 rather than treating upon school entry at age 6.

Methods: The most notable feature of this head-to-head study is the lack of a control group. Understanding it would be unethical to fail to treat egg-positive children, uninfected children could have been included and followed for initial and final hepatomegaly. Without that, it’s not clear to me how authors can conclude that the improvements seen are the result of treatment rather than e.g., regression to the mean or even seasonality of other causes of hepatomegaly like subclinical malaria. Unless I’m missing something, this needs to be acknowledged as a major limitation in the Discussion, including considering whether to recommend a followup study with uninfected controls before reaching conclusions about effectiveness of treatment.

Methods: please provide a citation for the use of a factor of 24 to convert eggs per slide to eggs per gram

Methods: not necessarily a weakness, but I’m curious as to why authors did not choose to collect a followup stool sample to test the impact of treatment on stool positivity or epg. Would briefly address your reasoning here or in the Discussion.

Methods: Similarly, I’m curious as to why authors assessed stool egg density on enrollment, but chose not to do anything with that data except descriptive analysis comparing followed children with those lost to followup. To me, the obvious next step would have been to stratify children by initial infection density when analyzing for hepatomegaly regression with treatment. And given the uncontrolled nature of the study, a pattern showing more improvement in more heavily infected children could help bolster your case that the improvement seen did in fact result from treatment. Or if authors intend to publish this separately instead, would just say so.

Discussion: similar to the comment on the introduction, it seems to me a reader might wonder: “The only thing that improved with treatment in these kids was hepatomegaly, not indicators of periportal fibrosis. Why should I assume this will improve their long-term liver function, as compared to just treating them at age 6?” Authors should either address this or acknowledge the uncertainty and perhaps argue for erring on the side of caution by treating early (as we do by treating school-aged kids rather than just adults), since it would be difficult to conduct an ethical study to answer this definitively. It might also help to briefly clarify whether or not liver inflammatory markers (AST, ALT) have a prognostic role here and should be investigated.

Reviewer #3: This article is a well-written description of a study to compare morbidity measures among pre-school aged children (12-60 months) and describes the effect of either a single or double dose of praziquantel on regression of identified morbidity. There is relatively little information available in the published literature regarding morbidity in this age group, and even less with regard to morbidity due to schistosomiasis and response to treatment. This manuscript will contribute much to the general knowledge available on this topic. Based on the findings presented here, pre-school aged children do experience morbidity due to schistosomiasis, which can be alleviated with treatment, and they should be included in mass-treatment campaigns.

The population sampled here has been well-described in previous manuscripts by these same authors. They used Kato-Katz methods for detection of schistosome ova in stool, testing three separate stools collected on different days and preparing multiple slides from each stool. This is in accordance with best practices as described by WHO. Kato-Katz can have lower sensitivity in areas with low-endemicity; however, the area studied is a high-endemic area of schistosomiasis along Lake Victoria, thus this method should adequately detect infection.

It is unfortunate that malaria status was not investigated in this study. In areas where the two infections are co-endemic it is necessary to understand the attributable morbidity of each in the population in order to more fully understand the likely effects of control measures for schistosomiasis, especially for morbidity markers like spleen size or anemia. With regards to abdominal ultrasound, only spleen and liver size are described here. Additional important measurements in identifying schistosomiasis-related morbidity are portal vein dilatation and periportal vein thickening - why were these measurements not included? Consider explaining in the Discussion section.

Regardless, I believe this paper should be accepted for publication. There are some additional minor points listed below that would help strengthen the paper further:

1. Introduction, lines 75-77: Consider finding other references than the ones currently listed (4 and 5), which are supposed to support the idea that infection and morbidity attributable to S. mansoni infection was thought to be rare among pre-school aged children. However, reference #4 (King 2006) actually described S. hematobium, not S. mansoni, and reference #5 (King 2010) does not describe the prevalence of morbidity among pre-school aged children or whether or not it is rare.

2. Materials and methods, Measures: The references included describe only Kato-Katz stool testing or abdominal ultrasound, but not hemoglobin. Consider including a reference or a short description of hemoglobin testing methods and estimation procedures.

3. Materials and methods, Anaemia assessment and abdominal ultrasound examination, line 122: The reference here for height-related data from a healthy Senegalese community (reference #30), is the same as reference #26.

4. Materials and methods, Anaemia assessment and abdominal ultrasound examination, line 123: Consider including a table or chart (perhaps in Table 2?) with the reference values for the liver and spleen measurements used in this study. This would provide greater context for the reader.

5. Materials and methods, Anaemia assessment and abdominal ultrasound examination: Consider also including a description of the ultrasound equipment (e.g. manufacturer and type of probe) and procedures used in this study (e.g. how many sonographers, were the sonographers blinded to the patient’s treatment arm, was there an assessment of inter-observer and intra-observer variance, etc) or provide a reference if it has been previously described in another publication. Again, this would provide greater context for the reader, and is an important point for ultrasound studies.

6. Results, Figure 1: Were the morbidity measurements at baseline similar for those who were retained in the study compared to those lost to follow-up? It would be important to know if those who were lost to follow-up had greater morbidity at baseline than those who were retained in the study.

7. Results, Table 2: This table compares the pre and post-treatment organometric measurements within the single and double praziquantel treatment groups. Maybe I missed it somewhere in the text, but were the pre-treatment baseline measurements similar between the two treatment groups? Also, the reference ranges used in this study (see comment #4) could be included in this table.

8. Discussion, last paragraph: This describes the interpretation of the increase in hemoglobin after treatment. The authors state that it is difficult to interpret, and that 9 children who were found to be severely anemic also received iron tablets in addition to their treatment which may have contributed to the increase in hemoglobin. While this is likely true, it would only be true for those 9 children, which is a relatively small proportion of the overall sample. In Nalugwa et al., (2017) (which I believe is the same study population described here?) anemia was significantly associated with intensity of schistosomiasis infection. Here, in the remainder of the children who experienced resolution of their anemia without iron supplementation, that improvement could be attributable to treatment for their S. mansoni infection. This is an important finding; but, because these children’s malaria infection status is unknown and the area is known to be co-endemic for malaria, it should still be interpreted with some caution.

9. References: Reference #26 and #30 are the same. Also, check the numbering – there is a blank reference #21 in between #27 and #28. Reference #38 appears to be incomplete – I believe this is a book chapter(?), but the book and chapter information are missing.

Reviewer #4: The paper seems to be continuation of a previous clinical trial already published (PLoS Negl Trop Dis. 2015;9(5):e0003796), so that point should be stated clearly somewhere.

Through both papers it is quite clear that beyond those studies there is a big logistical investment for carrying such research work on the “field”.

My major concern relies on the design of the study.

The fact of that work be the continuation of a previous trial is because methodologically is crucial. Authors refers to the sample size calculation to the previous study, where the number of participants were estimated according to another different outcome (cure), so what authors are doing is following the previous cohorts. Therefore, strictly it is not a randomized clinical trial.

That could have an important impact in the measure of the intervention effect (mainly due to the lower expected numbers of patients who complete the follow up period).

On the methods section, authors say that stool parasitological stool examination would be performed, but there are no results regarding that issue.

One of the variables that are measured is hemoglobin. Analyzing the results, probably, is one of the parameters that have been more affected by the intervention. Unfortunately for the study, it is stated in the discussion section that patients received albendazole. That information should be added at the methods.

Considering those facts what it could be said is that authors could not demonstrate if there is any difference between one or two doses, basically because the study was not designed for that.

Neither can it be said that: “…any of doses of praziquantel are effective in reversing morbidity attributable to S. mansoni..” What it has been improved are only a few of the several parameters identified by authors as related to the effect of S. mansoni (being one of them the Hb, but confounded with the effect of albendazol)

In summary my opinion is that the paper could be published but need to be rewritten taking into account the important methodological limitations and being more realistic when obtaining conclusions.

As a final minor comment, table 3 should be expressed as table 2 (same columns….)

Reviewer #5: This paper addresses a very important issue, the consequences of treating schistosomiasis in small children, who are mostly excluded from MDA programmes. In particular, it analyses the regression of severe injuries produced by S. mansoni at the abdominal level and observes an important but not complete regression of these injuries. Furthermore, no difference is observed between single and double dose treatment.

Together, these data are very valuable to both clinicians and policy makers in endemic areas, and increase evidence of the importance of treatment from the early ages of life

Major comments

The discussion in general does not go into much depth in several aspects. While the study is valuable and provides interesting information, it has some limitations and potential biases that should be detailed. A more thorough review of the literature should be advised.

Minor comments :

Methods

The blinding of the study should be better explained. We may think that the study was not blinded, as the first treatment is the same in the two groups and no placebo is mentioned in the second dose

Results

Table 2: the p value should be mentioned instead of “non-significant” value

Discussion

An important finding of the study is the high proportion of pathology detected in this group of children under six years old, both at the abdominal level and taking into account the anemia. A discussion of this issue, commenting on the literature on the importance of the pathology at this early age would give even more weight to advocacy for treating children in this age group.

Within the limitations, it would be necessary to comment on the absence of results of parasitological control and intensity of the infection, since these data would allow us to analyze the relationship of the lesions with the decrease of the parasitic load. As well, there are no data on adverse effects, which are one of the theoretical problems to exclude this age group from the control programs.

Another limitation is the absence of a control group, as the dynamics of injuries in this age group are unknown. This can easily be justified on ethical grounds, but would allow further discussion on the dynamics of injuries.

Line 231 : The discussion of anemia should also be deepened, with more reference to the relationship between schistosome infection and anemia after treatment

The last paragraph would have to be specified and detailed. The patients who were severely anemic were equally distributed between both groups? Was the albendazole given systematically to all patients? 5% of the patients who were infected with hookworm were equally distributed in all groups?

Conclusions

Line 240. The claim that anemia was reduced by treatment with praziquantel cannot be taken out of the study, as it is considered difficult to interpret in the discussion.

Linea 245. The administration of one dose of praziquantel is recommended, based on the logistical facility with respect to the administration of two doses, but it should be specified or deepened in the recommendation: is it proposed at any age? annually? Only once before six years?

Limitations

As previously explained, many more limitations and potential bias should be adressed.

Line 250: “Nevertheless, we had enough numbers at 8 months to give us adequate power to (>0.8) compare changes in liver size assuming a difference of 12% for single dose compared to double dose. This data should be placed in the results section, not in the conclusions.

6. PLOS authors have the option to publish the peer review history of their article (what does this mean?). If published, this will include your full peer review and any attached files.

Reviewer #1: No

Reviewer #2: No

Reviewer #3: No

Reviewer #4: No

Reviewer #5: No

---

## [Author Response · Author response to Decision Letter 0]

7 Sep 2020

Academic editor comments:

i. Please ensure that your manuscript meets PLOS ONE's style requirements, including those for file naming. 

Response: Followed the online PLOSOne formatting sample for guidance.

ii. PLOS ONE requires that methods are described in enough detail to allow suitably skilled investigators to fully replicate the study. 

Response: More detail for assessment of anaemia and examination of infections is described as advised. The treatment with praziquantel is also described further.

iii. Please also provide a sample size calculation. 

Response: This paper is a continuation of a previous clinical trial; details of study methods are described in these reports; lines 101-102.

iv. Thank you for stating in your ethics statement "Only those children who had written informed consent from their caregivers were included." 

a. Response: This has been stated in the ethics review as “Only those children who had written informed consent from their caregivers were included in this study.” Lines 162 – 163.

b. The statement “The trial was registered with ClinicalTrial.gov, identifier: NCT01901484" is now included in the ethics statement.

v. Finally, we note that your study was registered on May 30, 2013. Please add this information to the text of your manuscript. In addition, please also provide the date range in which this study took place in the text of your manuscript. 

Response: This study is registered with ClinicalTrial.gov and took place from June 2013 to February, 2014. Lines 106-107.

Reviewer #1:

i. a. Why are preschool age children not typically included in mass drug administration (MDA) campaigns? 

Until recently, it was thought that younger children don’t develop complications of chronic schistosomiasis and therefore don’t have to be treated. There is lack of child friendly treatment for children; the current praziquantel tablet is big and bitter. Lines 80 - 88.

b. Were the children treated with whole or crushed pills?

The children in this study were treated with crushed pills. Explained in lines 137-140.

ii. a. Describe the prevalence and interventions of malaria in the study region, its typical yearly timing in relation to the baseline and follow up of this study.

The prevalence of malaria and interventions that were conducted during the period that this study was carried out is described in lines 285 – 288. 

b. Is schistosomiasis liver fibrosis associated with liver image pattern C or B?

In this study, liver image pattern C, indicating peripheral rings and prominent pipe stems is described and considered to be schistosomiasis-associated fibrosis. Lines 232 – 233. 

Reviewer #2: 

i. While randomized, I don’t believe this can be considered a controlled trial since there is no control (untreated) group. Consider something like “a randomized dosing strategy trial” or “a randomized trial of one- vs. two-dose treatment”. 

The study title has changed to “Regression of Schistosoma mansoni associated morbidity among Ugandan preschool children following praziquantel treatment: A randomised trial” with word controlled deleted since there was no control group in this study. 

ii. Clear the confusion in language with regard to claims about morbidity. 

The entire introduction corrected and rewritten. Lines 75 – 97.

iii. Clarify the age groups that are being currently excluded from MDA in Uganda. 

Preschool children are maintained in the text; children who are young and not in school are not included in the national schistosomiasis MDA, in Uganda.

iv. Without a control group the authors should have included uninfected children and followed for initial and final hepatomegaly.

This paper is a continuation of a previous clinical trial and a group of uninfected children was included as discussed in lines 241 – 243. details of study methods described in these reports. The children were their own controls as explained in lines 279 – 282.

v. Please provide a citation for the use of a factor of 24 to convert eggs per slide to eggs per gram

Citation for the use of a factor of 24 to convert eggs per slide to eggs per gram provided. Line 126 reference number 27

Reviewer #3: 

i. Why did the authors not include measurements of portal vein dilatation and periportal vein thickening to identify schistosomiasis related morbidity?

Measurements of portal vein dilatation and periportal vein thickening are not reported in this study because all children had normal sizes of hepatic portal vein at baseline and at follow-up. Lines 183 – 185. 

ii. Consider finding other references than the ones currently listed (4 and 5)

References King 2006and King 2010 deleted and relevant ones cited.

iii. A short description of hemoglobin testing methods and estimation procedures should be included in the methods.

Hemoglobin testing methods and estimation procedures included. Lines 119 – 123

iv. The reference for height-related data from a healthy Senegalese community should be put rightly. 

Reference corrected and changed. Line 123 refence number 25.

v. Reference values for the liver and spleen measurements should be included in Table 2.

Reference values for the liver and spleen measurements used in this study described in Table 2

vi. Please give a detailed description of materials and methods for anaemia assessment and abdominal ultrasound examination.

This is fully described in other reports. Lines 101 – 102 

vii. Described the association of anemia with intensity of schistosomiasis infection.

Heamoglobin level among children increased with decrease in intensity of S. mansoni infection. Discussed in line 2587-265

Reviewer #4: 

i. State whether this paper is a continuation of a previous clinical trial already published (PLoS Negl Trop Dis. 2015;9(5): e0003796). 

It is clearly stated in line 101-102 that this paper is a continuation of previous reports. References number 3, 9 and 17. 

ii. This study it should not be considered a randomized clinical trial.

Although the study did not have a control group it remains a randomized trial as changed in the study title: “Regression of Schistosoma mansoni associated morbidity among Ugandan preschool children following praziquantel treatment: A randomized trial” . 

iii. It should be stated in the methods that children received albendazole.

It has been indicated that children received albendazole at baseline. Line 142-143

Reviewer #5: 

i. The discussion needs thorough review of more literature.

More literature has been cited in the discussion. Line 222 – 265

ii. It should be explained whether the study involved blinding.

The nurses were blinded to the treatment groups. Lines 140-141

iii. In Table 2: the p values are stated as ns if not significant at P > 0.05.

iv. Discuss the schistosomiasis pathology in preschool children

Schistosoma mansoni related pathology discussed further. Lines 232-236 

v. Were there any adverse effects following chemotherapy?

There were no reports of adverse effects among the children following treatment with PZQ. Lines 144-148

vi. Discuss in detail the relationship between schistosome infection and anemia after treatment.

The relationship between schistosome infection and anemia after treatment is discussed in detail. Lines 257-265

---

## [Decision Letter · Decision Letter 1]

4 Dec 2020

PONE-D-20-12223R1

Regression of Schistosoma mansoni- associated morbidity among Ugandan preschool children following praziquantel treatment: A randomized trial

PLOS ONE

Dear Dr. Nalugwa,

Thank you for submitting your manuscript to PLOS ONE. After careful consideration, we feel that it has merit but does not fully meet PLOS ONE’s publication criteria as it currently stands. Therefore, we invite you to submit a revised version of the manuscript that addresses the points raised during the review process.

Please respond to ALL reviewer comments, as some of the original reviewers did not feel that their concerns and comments had been adequately addressed by the authors.

We look forward to receiving your revised manuscript.

Kind regards,

David Joseph Diemert, M.D.

Academic Editor

PLOS ONE

Reviewers' comments:

Reviewer's Responses to Questions

**Comments to the Author**

1. If the authors have adequately addressed your comments raised in a previous round of review and you feel that this manuscript is now acceptable for publication, you may indicate that here to bypass the “Comments to the Author” section, enter your conflict of interest statement in the “Confidential to Editor” section, and submit your "Accept" recommendation.

Reviewer #1: (No Response)

Reviewer #2: (No Response)

Reviewer #4: (No Response)

Reviewer #5: (No Response)

Reviewer #6: (No Response)

2. Is the manuscript technically sound, and do the data support the conclusions?

Reviewer #1: Yes

Reviewer #2: Partly

Reviewer #4: Partly

Reviewer #5: Partly

Reviewer #6: Yes

3. Has the statistical analysis been performed appropriately and rigorously? 

Reviewer #1: Yes

Reviewer #2: Yes

Reviewer #4: No

Reviewer #5: Yes

Reviewer #6: No

4. Have the authors made all data underlying the findings in their manuscript fully available?

Reviewer #1: Yes

Reviewer #2: Yes

Reviewer #4: Yes

Reviewer #5: Yes

Reviewer #6: Yes

5. Is the manuscript presented in an intelligible fashion and written in standard English?

Reviewer #1: Yes

Reviewer #2: Yes

Reviewer #4: Yes

Reviewer #5: Yes

Reviewer #6: Yes

6. Review Comments to the Author

Reviewer #1: The authors have responded to most of the reviewers' comments. However, the response to the question about malaria prevalence and interventions is not in lines 285-288 as the authors claim. Maybe the line numbers are off but this is of critical importance in considerations about anemia. They do semi-respond in the section on limitations of the study but overall they could be more forthcoming about the possibility that the changes in anemia in the study participants could be confounded by malaria.

Other minor corrections:

--line 79 should be a less than or equal to sign rather than a greater than or equal to sign

--line 131 is only the magnification of the objective lens, when coupled with the ocular lenses, the overall magnification is 100X

--line 211 says children who had liver pathology (defined by the authors as Image Pattern C) were more likely to be lost to follow up. This is not true as the p value is 0.76.

Reviewer #2: General: In some cases authors have responded satisfactorily to my initial comments, however, the point-by-point reply was a highly compressed summary in which many comments were not addressed, and the manuscript revisions also have not addressed them. I have copied these from my original review below, with slight editing in some cases.

Introduction: Currently, authors are primarily using the introduction to make the case that preschool-aged children should be treated at all, but then the study focuses on frequency of treatment instead. We need additional text to make the case to the reader of why looking at 2 vs 1 dose adds value. I.e., was there data or concern that younger kids were likely to require a double dose and thus would not benefit from inclusion in MDAs (which to my knowledge are only single-dose)?

Methods: not necessarily a weakness, but I’m curious as to why authors did not choose to collect a followup stool sample to test the impact of treatment on stool positivity or epg. Would briefly address your reasoning here or in the Discussion.

Methods: Similarly, I’m curious as to why authors assessed stool egg density on enrollment, but chose not to do anything with that data except descriptive analysis comparing followed children with those lost to followup. To me, the obvious next step would have been to stratify children by initial infection density when analyzing for hepatomegaly regression with treatment. And given the uncontrolled nature of the study, a pattern showing more improvement in more heavily infected children could help bolster your case that the improvement seen did in fact result from treatment. Or if authors intend to publish this separately instead, would just say so.

In the discussion: What are the implications of the above issues [around logistical challenges of reaching out-of-school children] for your recommendations…?

Finally, a few new comments based on the revised Discussion text:

-Discussion: Please strengthen your argument by briefly assessing alternative explanations for the improvements in hepatomegaly seen with treatment. To me the most obvious would be if malaria was responsible for some of the initial hepatomegaly, and local seasonality patterns are such that community infection prevalence is lower around the time of the followup testing. (The brief mention of malaria seasonality does not address this.) Another possibility which is important in some SSA regions is aflatoxin exposure depending on the local staple food and storage practices.

- Discussion line 269: this is the first reference I’ve seen to a cure rate being determined in your study. If these children were in fact followed up with repeat stool testing, at minimum a brief mention of the findings and a reference to where they are more fully presented is needed.

- Discussion lines 277-281: It appears you are largely basing the argument that treatment earlier in childhood may decrease clinical morbidity (liver fibrosis and failure) in adulthood on the full resolution seen in your few cases diagnosed with periportal fibrosis. 1) are you sure that it’s appropriate to interpret these sonographic findings in children using fibrosis criteria developed for adults, when the reversibility itself may suggest they were not true fibrosis? 2) to make this argument you would need to briefly provide comparisons – did the other studies in children show the same findings? In contrast, have studies in adults shown that these findings do not resolve with treatment?

Reviewer #4: Unfortunately my comments were not addressed

Reviewer #5: Not all reviewers' comments have been answered

The lack of a control group cannot be explained. The children cannot be their own controls, as we cannot know how the natural evolution of a hepatomegaly would be, for example, in a non-infected child, since we cannot rule out the possibility that it may resolve spontaneously, be related to the seasonality of malaria, other conditions, etc. Therefore, this has to be clearly specified as a major limitation of the study. Although the section on limitations contains these considerations, it immediately considers them as not relevant, when they should be considered as possible and relevant biases.

The text specifies that the treatment was blind for the nurses, but it does not explain how this was done, a placebo was used for the second dose ? How was it adressed at that time?

The limitation on the absence of results of parasitological control and the intensity of the infection has not been answered

Reviewer #6: Abstract:

Line 52 - you probably mean that you took haemoglobin measurements, as 'anaemia' cannot really be measured directly.

Line 55 - you probably mean that there was no difference between the changes in proportion of children with enlarged liver, because that is what your outcome measure is; it is not 'liver size' per se as implied by this sentence. In both line 53 and 54 you should include an estimates of the differences between those two pairs of proportions with the 95% confidence intervals (and p-values if you wish). Same in line 57 and line 60.

Line 60 - the conclusion that these results suggests that one dose is as effective as two doses is incorrect given the study design (that would have been correct if this was a non-inferiority trial); the more appropriate conclusion is that there is no evidence of a difference in effect between the two doses.

Introduction:

Line 84 - check that you really mean that the tablet is 'sour' (like lime) and not 'bitter' (like quinine). You refer to bitterness in your response to the previous round of reviews and also in line 170.

Methods:

Line 113 - what you mean is the 5% level of significance, not the 95% level.

Also, this description of the sample size calculation is incomplete, as it should explicitly state the outcome upon which it was based (morbidity is generic) and also state the proportion of this outcome in at least one treatment group (usually the comparison/baseline/control group).

The McNemar-Bowker test is not appropriate for these data because individuals in these data are not paired. You should instead conduct a regression analysis such as binomial regression to estimate proportion (risk) differences with their confidence intervals (e.g. 'binreg' in Stata with the 'rd' option for risk differences). Furthermore, a comparison of baseline characteristics between groups is not appropriate for a randomised group assignment as the groups are expected to be balanced and any observed differences are likely to be due to chance.

Results:

Table 1 should describe the baseline characteristics of the children retained in the study only, comparing between those assigned to the two doses. However, there should be no statistical tests comparing the groups. The current Table 1 is also useful as-is, but should be moved to an appendix, with a comment in the main text (e.g. what's currently stated in lines 202 to 203) describing how the retained sample and the lost-to-followup groups compare. The focus of the first paragraph should be on describing the retained sample as shown in this new table 1 which I am suggesting to be included. In this table counts and proportions of categorical variables, and means and SDs or medians and IQRs of continuous variables should be reported - remember, no statistical tests comparing the two dosage arms.

The description of mordibity in line 213 belongs to the methods.

The analysis and presentation of results in table 2 is inappropriate. I would advice that you define the outcomes as binary variables (i.e. enlarged vs not) and then fit a model estimating risk differences for enlargement pre-vs-post in the two treatment groups. To do this you would first format the data such that each row represents one observation per child per timepoint, meaning each child would appear twice in the data: once for the pre- and once for the post- treatment times. You would need a variable indicating timepoint (e.g. 0 for pre-treatment and 1 for post-treatment). You would also have a variable for treatment group i.e. single vs double dose. Then fit a model with each binary outcome and with interaction between treatment group; you can subsequently include covariates if you also want adjusted estimates. For example in Stata: binreg outcome treatment_group##time, rd (you can then repeat this with any adjusting variables). In this model, the effect of the treatment_group#time interaction, estimated as a risk difference, is the difference in pre-vs-post treatment risks of organ enlargement between the groups. This is what you should then report for each outcome in Table 2 as the treatment effect with 95% confidence interval and p-value. You may report both unadjusted and adjusted effects. Please consult a statistician for advice on this. You can do the same for anaemia (as a binary variable) or fit a linear regression model with the same interaction above with continuous haemoglobin. Each results table should include the count and proportion of the outcome pre-treatment and post-treatment in each arm (or the mean and SE for continuous outcomes) and the treatment effect estimated as I have described, with the 95%CI and p-value.

Discussion: given my concerns about the analysis I cannot comment on the rest of the manuscript.

7. PLOS authors have the option to publish the peer review history of their article (what does this mean?). If published, this will include your full peer review and any attached files.

Reviewer #1: No

Reviewer #2: No

Reviewer #4: No

Reviewer #5: No

Reviewer #6: No

---

## [Author Response · Author response to Decision Letter 1]

7 Feb 2021

Response to Reviewers letter has been uploaded.

---

## [Decision Letter · Decision Letter 2]

22 Mar 2021

PONE-D-20-12223R2

Regression of Schistosoma mansoni- associated morbidity among Ugandan preschool children following praziquantel treatment: A randomized trial

PLOS ONE

Dear Dr. Nalugwa,

Thank you for submitting your manuscript to PLOS ONE. After careful consideration, we feel that it has merit but does not fully meet PLOS ONE’s publication criteria as it currently stands. Therefore, we invite you to submit a revised version of the manuscript that addresses all the points raised during the review process.

We look forward to receiving your revised manuscript.

Kind regards,

David Joseph Diemert, M.D.

Academic Editor

PLOS ONE

Reviewers' comments:

Reviewer's Responses to Questions

**Comments to the Author**

1. If the authors have adequately addressed your comments raised in a previous round of review and you feel that this manuscript is now acceptable for publication, you may indicate that here to bypass the “Comments to the Author” section, enter your conflict of interest statement in the “Confidential to Editor” section, and submit your "Accept" recommendation.

Reviewer #1: All comments have been addressed

Reviewer #2: (No Response)

Reviewer #5: All comments have been addressed

Reviewer #6: (No Response)

2. Is the manuscript technically sound, and do the data support the conclusions?

Reviewer #1: Yes

Reviewer #2: Partly

Reviewer #5: Yes

Reviewer #6: Yes

3. Has the statistical analysis been performed appropriately and rigorously? 

Reviewer #1: Yes

Reviewer #2: Yes

Reviewer #5: Yes

Reviewer #6: No

4. Have the authors made all data underlying the findings in their manuscript fully available?

Reviewer #1: Yes

Reviewer #2: No

Reviewer #5: Yes

Reviewer #6: Yes

5. Is the manuscript presented in an intelligible fashion and written in standard English?

Reviewer #1: Yes

Reviewer #2: Yes

Reviewer #5: Yes

Reviewer #6: Yes

6. Review Comments to the Author

Reviewer #1: (No Response)

Reviewer #2: Introduction: This is a restatement of the comment on the Introduction in my last review which reviewers noted they did not understand. There is still a crucial disconnect between the way the authors frame the study’s contributions and what the study design can actually show. Most of the introduction is about the exclusion of PSAC from praziquantel MDAs altogether, and the authors situate their findings as “helpful in guiding policy and practice decisions regarding the routine treatment of this important age group”. But then the study design looks at morbidity regression with 1 vs 2 doses. Demonstrating that children getting 1 dose had equivalent medium-term improvement those getting to 2 does not demonstrate that treatment caused those outcomes and is superior to no treatment. Though this issue risks causing important confusion to readers, it may be a purely semantic problem that could be easily addressed by reframing to state that this study was intended to determine two things: 1) whether morbidities regressed with treatment at all over time, and 2) if so, whether one dose was equivalent to two since it would be simpler to achieve. Currently the introduction conflates these two concepts.

Discussion: The authors continue to frame the results as clearly showing that PZQ treatment was effective in causing the morbidity reductions seen in their study. The study design does not allow making this determination and findings should not be described as representing effectiveness at all.

The authors have rejected suggestions in previous comments to nuance this claim by providing alternate explanations for the changes seen (e.g. malaria – which is discussed extensively as an explanation for findings from other studies, but not as a possible cause of spurious association with treatment for their own findings.) But even these comments would only have helped in the context of a larger reframing that has not been done: acknowledging that an uncontrolled pre-post study cannot prove causality, and at best can only suggest an association worth investigating further. Without an uninfected (not untreated) control group followed on the same outcomes over the same time period, we cannot rule out changes in malaria prevalence, other exposure, or even natural changes with age, as the cause of improvement in findings. (E.g. is organomegaly more prevalent in younger children in general than older ones in this population?) (The Kenyan schisosomiasis study referenced in the reply also followed an uninfected comparison group over time.) Fundamentally, the claims for the type and strength of evidence this provides need to be dialed back and the authors need to proactively point out the remaining evidence gaps. But authors can still justly claim that their study provides longer-term followup data than was previously available in PSAC and will help guide dosing if/when programs begin to target PSAC.

Reviewer #5: Most of the issues have now been adressed and corrected. I think the article is suitable for publication, I have two minor comments at the moment.

Line 317 : “Consequently, our results should be 318 generalizable to children less than six years infected with S. mansoni”. This statement is difficult to maintain in a limitations section, without a control group.

Line 322: “Besides, the major control strategy for malaria in the study area is with use of insecticide treated mosquito nets with a very high household coverage of > 90%”. In the same way, this statement is not justified in a limitation section

Reviewer #6: Major comments:

The authors have presented a rebuttal to my previous suggestion of the appropriate approach to analysis of these data. I am not convinced by the rebuttal for the following reasons:

(1) the study compared a single vs. double dose of praziquantel on its effect on regression of morbidity as measured by the sizes of their spleens and livers as well as liver fibrosis. The authors cite the proportions with spleen enlargement, liver enlargement and liver fibrosis (all binary outcomes) in the two treatment groups but yet in the methods and in Table 2, these outcomes are presented as three-level categorical variables. Perhaps this was not the authors original intention and this has been introduced during peer review. However, one reason why I am not convinced by the authors rebuttal that they were interested in liver size as a 3-level spectrum is that the sample size calculation is based on a binary outcome! The way an outcome is specified for the sample size is usually the way it is specified in the main analysis. In any case, it would have been statistically even more powerful to use the actual liver/spleen size eg. in centimetres as the outcome of the analysis rather than categorising it to 3-levels, if the worry is about blurring that aspect of morbidity as the authors indicate. Alternatively, there is the option to use an ordinal logistic regression to model the 3-level ordered categorical outcome and adjusting for the baseline level of the outcome, with the treatment indicator as the main explanatory variable.

(2) assuming the outcomes are in fact 3-level categorical variables, in table 2, you present p-values comparing each level of each 3-level categorical variable in the pre- vs post- treatment period, repeating this for the single dose and double dose arms. This is inappropriate because firstly, these three categories are not independent of each other, since belonging to one category necessarily means you don't belong to the other two categories. Secondly, comparing the pre- and post-treatment groups in each of the arms separately does not tell us anything about how the arms compare to each other, which is the aim of the study! These p-values are therefore meaningless in as far as evaluating evidence for the difference between the two treatments is concerned.

The authors assert that McNemar-Bowker test is appropriate in this setting because observations are paired in the sense that there are pre-treatment and post-treatment in the same individuals. This is a misinterpretation of the concept of pairing in as far as exploring the causal effect of treatment is concerned. In this setting a pre-treatment value of an outcome is a 'baseline' and has not been affected by treatment; it is a value to be adjusted for in the analysis. Only the post-treatment values of the outcome are relevant for the comparison of the treatment groups as they are the only values expected to have been causally influenced by treatment. Hence my previous suggestion to fit a model which estimates the difference in the post-versus-pre-treatment values of the outcome in the one dose versus two dose groups i.e. a difference-in-difference analysis (or alternatively as discussed above linear regression on liver and splenic sizes in cm and even haemoglobin post treatment, or ordinal logistic regression on 3-level categorical measures, adjusting for the pre-treatment values with the one vs. two dose variable as the main explanatory variable, with or without adjustment for other factors so desired).

This is a good study and I think that if the authors could improve the statistical analysis as advised, perhaps consulting with a statistician as well, I think it would be a great paper. The authors should also look at the CONSORT guidelines to guide them on how and what to report in each section of the manuscript. I sympathise with the authors having to deal with multiple review comments, and as far as I can see, there is no contradiction in reviewers' feedback as far as the statistical analysis and reporting is concerned.

Minor comments:

In the response document you indicate that you have changed the description of the pill from 'sour' to 'bitter' but the text still says sour.

I had suggested previously that the difference-in-difference of the pre- vs post-treatment prevalences in the two groups be presented, with confidence intervals (and p-values if desired) but this has not been done. Without this, it is not possible to evaluate the conclusion that there is no evidence of a difference in effect - the reader needs to see this difference in effect with its confidence intervals and note whether it does or does not exclude no difference.

I also note that parameters for a sample size calculation are now presented i.e. assuming a difference of at least 12% and assuming 75% prevalence of the primary outcome in the two-dose group. It is not enough to present these parameters without justification e.g. some citation showing that these would be the expected values or why a 12% difference is clinically important. The 'sample size' section should be moved to just above the statistical analysis section.

7. PLOS authors have the option to publish the peer review history of their article (what does this mean?). If published, this will include your full peer review and any attached files.

Reviewer #1: No

Reviewer #2: No

Reviewer #5: No

Reviewer #6: No

---

## [Author Response · Author response to Decision Letter 2]

2 May 2021

All comments from reviewers number 2, 5 and 6 have been attended to as attached in the rebuttal letter.

---

## [Decision Letter · Decision Letter 3]

27 May 2021

PONE-D-20-12223R3

Regression of Schistosoma mansoni- associated morbidity among Ugandan preschool children following praziquantel treatment: A randomized trial

PLOS ONE

Dear Dr. Nalugwa,

Thank you for submitting your manuscript to PLOS ONE. After careful consideration, we feel that it has merit but does not fully meet PLOS ONE’s publication criteria as it currently stands. Therefore, we invite you to submit a revised version of the manuscript that addresses the points raised during the review process.

Please respond to the previously noted comment by Reviewer #2 regarding the conclusions and discussion not being justified by the results.

We look forward to receiving your revised manuscript.

Kind regards,

David Joseph Diemert, M.D.

Academic Editor

PLOS ONE

Journal Requirements:

Reviewers' comments:

Reviewer's Responses to Questions

**Comments to the Author**

1. If the authors have adequately addressed your comments raised in a previous round of review and you feel that this manuscript is now acceptable for publication, you may indicate that here to bypass the “Comments to the Author” section, enter your conflict of interest statement in the “Confidential to Editor” section, and submit your "Accept" recommendation.

Reviewer #2: (No Response)

2. Is the manuscript technically sound, and do the data support the conclusions?

Reviewer #2: Partly

3. Has the statistical analysis been performed appropriately and rigorously? 

Reviewer #2: I Don't Know

4. Have the authors made all data underlying the findings in their manuscript fully available?

Reviewer #2: Yes

5. Is the manuscript presented in an intelligible fashion and written in standard English?

Reviewer #2: Yes

6. Review Comments to the Author

Reviewer #2: I am baffled as to why the authors are repeatedly refusing to state consistently and clearly in the manuscript a crucial interpretation point that they appear to agree with in their response letters: An uncontrolled pre-post study design cannot prove causality. Listing possible alternate explanations in the Discussion does not compensate for repeatedly and inappropriately using words like "effectiveness" and "effect" instead of "association". There is even the claim that MDA policy should be altered based on their findings, when the type of data presented does not meet the standard of evidence used for e.g. WHO policy recommendations. This is not a criticism of the study or the data, which in fact is valuable and should be considered within the body of evidence in policy updates.

The meaning of the rebuttal claim that comparing evolution of ultrasound abnormalities over time in infected treated vs uninfected PSAC cannot be done because 'results are different' is also unclear. However, even if there is a biomedical reason this comparison would be unhelpful, there remains the basic problem: we don't know whether the abnormalities initially seen in the enrolled children are 'worse' than would have been seen in uninfected children, or if the improvement over an 8-month period would have happened regardless of treatment. None of this is a problem with the study itself, but with the claims being made in the Discussion about what the study can prove.

7. PLOS authors have the option to publish the peer review history of their article (what does this mean?). If published, this will include your full peer review and any attached files.

Reviewer #2: No

---

## [Author Response · Author response to Decision Letter 3]

3 Jun 2021

Editor comment

Please upload a copy of Supporting Information Figure S2 which you refer to in your text on page 14.

Figure 1 was misplaced under Supporting Information as Figure S2 Fig 1 tiff and it has been deleted. Line 323 page 14

Figure 1 has been reuploaded under figure file.

Reviewer #2:

I am baffled as to why the authors are repeatedly refusing to state consistently and clearly in the manuscript a crucial interpretation point that they appear to agree with in their response letters: An uncontrolled pre-post study design cannot prove causality. 

We have had a big problem with these assertions. 

1. The aim of this study is not to establish/prove causality. Causality is already known. 

2. The data ought to be interpreted against this body of knowledge and not in isolation. 

3. It is unethical to have a control group that cannot be treated!!! 

Listing possible alternate explanations in the Discussion does not compensate for repeatedly and inappropriately using words like "effectiveness" and "effect" instead of "association". 

See comments above. Although it does not compensate it puts the study in perspective signaling that data should be interpreted with caution. However, by casting the study in available literature and known knowledge there is a higher likelihood that the observed changes are “causal”. Inconceivable that one can treat schistosomiasis to observe an “association”. 

There is even the claim that MDA policy should be altered based on their findings, when the type of data presented does not meet the standard of evidence used for e.g. WHO policy recommendations. This is not a criticism of the study or the data, which in fact is valuable and should be considered within the body of evidence in policy updates. 

The study adds to the body of knowledge suggesting that there is need for a change in policy. If children improve after treatment, they should be treated! 

The meaning of the rebuttal claim that comparing evolution of ultrasound abnormalities over time in infected treated vs uninfected PSAC cannot b done because 'results are different' is also unclear. However, even if there is a biomedical reason this comparison would be unhelpful, there remains the basic problem: we don't know whether the abnormalities initially seen in the enrolled children are 'worse' than would have been seen in uninfected children, or if the improvement over an 8-month period would have happened regardless of treatment. None of this is a problem with the study itself, but with the claims being made in the Discussion about what the study can prove. 

We think this came from a previous comment about the Kenyan study. This issue was actually raised by the reviewer previously (second review) (reference 6).

---

## [Decision Letter · Decision Letter 4]

19 Oct 2021

Regression of Schistosoma mansoni- associated morbidity among Ugandan preschool children following praziquantel treatment: A randomized trial

PONE-D-20-12223R4

Dear Dr. Nalugwa,

We’re pleased to inform you that your manuscript has been judged scientifically suitable for publication and will be formally accepted for publication once it meets all outstanding technical requirements.

Kind regards,

Matt A Price

Academic Editor

PLOS ONE

Additional Editor Comments (optional):

Reviewers' comments:

Reviewer's Responses to Questions

**Comments to the Author**

1. If the authors have adequately addressed your comments raised in a previous round of review and you feel that this manuscript is now acceptable for publication, you may indicate that here to bypass the “Comments to the Author” section, enter your conflict of interest statement in the “Confidential to Editor” section, and submit your "Accept" recommendation.

Reviewer #2: (No Response)

2. Is the manuscript technically sound, and do the data support the conclusions?

Reviewer #2: Partly

3. Has the statistical analysis been performed appropriately and rigorously? 

Reviewer #2: I Don't Know

4. Have the authors made all data underlying the findings in their manuscript fully available?

Reviewer #2: Yes

5. Is the manuscript presented in an intelligible fashion and written in standard English?

Reviewer #2: Yes

6. Review Comments to the Author

Reviewer #2: I'm sorry to see an apparent impasse in the revision of this paper, which would provide useful data to the international community.

However this manuscript proceeds, the authors may wish to consider an additional nuance: even if they consider it a previously settled point that PZQ treatment substantially improves organomegaly in preschool-aged children, this study design still cannot prove what fraction of the observed improvement is attributable to treatment vs other factors or natural regression over time. If authors feel that the overall body of research including their own data compels them to make strong claims about causality here, perhaps they will be more open to acknowledging that changes in organomegaly are potentially multifactorial and they have no way with this study design to determine what fraction of the improvement is due to treatment.

7. PLOS authors have the option to publish the peer review history of their article (what does this mean?). If published, this will include your full peer review and any attached files.

Reviewer #2: No

---

## [Editor Report · Acceptance letter]

5 Nov 2021

PONE-D-20-12223R4 

Regression of * Schistosoma mansoni * associated morbidity among Ugandan preschool children following praziquantel treatment: A randomised trial 

Dear Dr. Nalugwa:

I'm pleased to inform you that your manuscript has been deemed suitable for publication in PLOS ONE. Congratulations! Your manuscript is now with our production department. 

Kind regards, 

on behalf of

Dr. Matt A Price 

Academic Editor

PLOS ONE